# Social Frailty and Meaningful Activities among Community-Dwelling Older Adults with Heart Disease

**DOI:** 10.3390/ijerph192215167

**Published:** 2022-11-17

**Authors:** Yoshihiko Akasaki, Takayuki Tabira, Michio Maruta, Hyuma Makizako, Masaaki Miyata, Gwanghee Han, Yuriko Ikeda, Atsushi Nakamura, Suguru Shimokihara, Yuma Hidaka, Taishiro Kamasaki, Takuro Kubozono, Mitsuru Ohishi

**Affiliations:** 1Department of Rehabilitation, Tarumizu Central Hospital, 1-140 Kinko-cho, Tarumizu 891-2124, Japan; 2Graduate School of Health Sciences, Kagoshima University Faculty of Medicine, 8-35-1 Sakuragaoka, Kagoshima 890-8544, Japan; 3Department of Occupational Therapy, Nagasaki University Graduate School of Biomedical Sciences Health Sciences, 1-7-1 Sakamoto, Nagasaki 852-8520, Japan; 4Department of Occupational Therapy, School of Health Sciences at Fukuoka, International University of Health and Welfare, 137-1 Enokizu, Fukuoka 831-8501, Japan; 5National Institute for Minamata Disease, Ministry of the Environment, 4058-18 Hama, Kumamoto 867-0008, Japan; 6Doctoral Program of Clinical Neuropsychiatry, Graduate School of Health Science, Kagoshima University, 8-35-1 Sakuragaoka, Kagoshima 890-8544, Japan; 7Department of Rehabilitation, Medical Corporation, Sanshukai, Okatsu Hospital, 3-95 Masagohonmachi, Kagoshima 890-0067, Japan; 8Department of Cardiovascular Medicine Hypertension, Graduate School of Medical and Dental Sciences, Kagoshima University, 8-35-1 Sakuragaoka, Kagoshima 890-8520, Japan

**Keywords:** frailty, heart disease, older adult

## Abstract

Patients with heart disease are more likely to experience social frailty due to physical inactivity, which may affect meaningful activities such as hobbies. This study aimed to investigate (1) the association between heart disease and social frailty in community-dwelling older adults and (2) the characteristics of meaningful activities in community-dwelling older adults with heart disease. Data from 630 older adults who participated in a community-based health survey were obtained, including clinical history, meaningful activities, social frailty and psychosomatic functions. Participants were divided into two groups: those with heart disease (n = 79) and those without (n = 551), and comparisons were made. Social frailty was observed in 23.7% of participants with heart disease, and logistic regression revealed significant associations with heart disease and social frailty after adjusting for potential covariates (OR, 1.97; 95% CI, 1.06 3.67; *p* = 0.032). Participants with heart disease did not differ significantly in terms of satisfaction or performance; their frequency of engagement in meaningful activities was significantly lower than without heart disease (*p* = 0.041). These results suggest that heart disease and social frailty are associated in community-dwelling older adults, and that this demographic is inclined to engage in meaningful activities less frequently.

## 1. Introduction

In Japan, the incidence of heart diseases such as angina pectoris, heart failure, and valvular heart disease is low compared to that in western countries [1]. However, the rate has seen an increase in recent years, especially among older adults [2]. The Hisayama study found that the incidence of coronary artery disease has not changed much since 1950, although the mortality rate from coronary artery disease has decreased. This indicates that people with heart disease, such as coronary artery disease, are living longer [3]. Therefore, the development of adequate support for older adults living with heart disease is becoming a key issue.

There has been increasing recognition of frailty in the field of heart disease. Frailty is a state of decreased physiological reserve and resistance to stressors [4,5]. It is a multidimensional concept that incorporates physical, cognitive, emotional, and social elements [6,7,8,9]. Emma et al. followed heart failure patients and reported that they are at high risk of developing physical frailty [10]. These results show that exercise-related symptoms in patients with heart disease (e.g., shortness of breath or exercise induced chest pain) could lead to physical inactivity, making them more likely to become frail. Physical inactivity in heart disease patients may affect not only physical and mental functions but also social activity. As social activity frequently requires the integration of physical and mental capacities, social frailty may develop at a relatively early stage in the progressive trajectory of frailty [11]. However, the majority of reports on frailty in those with heart disease relate to physical frailty and cognitive functions, while the social domain of frailty has received little research attention.

While older adults with heart disease are encouraged to increase activity frequency, it has been reported that shortness of breath, fatigue, and the associated increase in rest periods may lead to a change to less demanding activities and a decrease in frequency [12,13]. These results suggest that older adults with heart disease may have difficulties in engaging in activities that are meaningful to the individual. Satisfying with meaningful activities in the older adults may be an effective support for social frailty [14]. Even in older adults with heart disease, regular participation in such activities has been shown to reduce the rate of social frailty [15]. However, to our knowledge, no studies have investigated the characteristics of meaningful activities that may reduce social frailty in patients with heart disease. For heart disease patients whose physical activity is limited, access to meaningful activities that take these physical limitations into account may promote social participation and contribute to the maintenance of physical and mental functions and the prevention of frailty.

Few studies have been conducted on social frailty in those with heart disease, and no studies have been conducted on meaningful activities to reduce social frailty in this demographic. Therefore, this study aims (1) to investigate the relationship between heart disease and social frailty in community-dwelling older adults and (2) to determine the characteristics of meaningful activities in community-dwelling older adults with heart disease. It is hoped that the findings of this study will contribute to the development of strategies that will facilitate social engagement among older adults with heart disease.

## 2. Materials and Methods

### 2.1. Participants

This cross-sectional study used data from the Tarumizu Study held during June 2019 to December 2019. It recruited participants aged 40 years or older, living in Tarumizu City at the time of the survey (during 2019) by mailing them reply cards. A total of 1028 participants were enrolled, and 690 of them aged 65 years or older and not requiring support or care according to the Japanese public long-term care insurance system were included this study. The exclusion criteria included: (1) history of dementia, cerebrovascular accidents, or depression, and (2) missing data on social frailty assessments, meaningful activity assessments or other measurements. After these exclusions, our final sample consisted of 630 community-dwelling older adults aged 65 years or older (mean age 74.2 ± 6.4 years, 62.9% female) (Figure 1). The ethics committee of the Faculty of Medicine, Kagoshima University approved the study protocol (Ref No. 170351, approval date: 26 October 2018), and informed consent was received from all participants before they were included in the study.

### 2.2. Measures

#### 2.2.1. Determination of Heart Disease

Heart disease data were obtained from the questionnaire, which asked about medical diagnoses that the participant had received from a doctor [16]. A current or previous diagnosis of heart disease was classified as either coronary heart disease or other heart disease (e.g., heart failure, aortic aneurysm, carotid arteries) based on the classifications used in previous research [17]. Of the 630 participants surveyed, 79 were found to have a current or past diagnosis of heart disease. This included angina pectoris (n = 46), myocardial infarction (n = 12), aortic aneurysm (n = 5), heart failure (n = 2), peripheral arterial disease (n = 4) and atrial fibrillation/flutter (n = 10). These 79 participants comprised the heart disease group with heart disease (mean age 76.3 ± 6.4 years, 48.1% female), and the remaining 551 participants comprised the ‘without heart disease’ group (mean age 73.9 ± 6.3 years, 65.1% female).

#### 2.2.2. Assessment of Social Frailty

In this study, social frailty is defined as “social domains of frailty,”, i.e., social characteristics or social factors that are often expressed in frailty individuals [18]. Social frailty was evaluated using a five-question measure developed by Makizako et al. [9]. These question measures are used in the field of heart disease [11,19]. The questions ask about aspects of social behavior and lifestyle associated with social frailty. These are as follows: living alone, going out less frequently compared with last year, not visiting friends sometimes, not feeling helpful to friends or family and not talking with someone every day. We divided the study population into two groups, that is, those who met two or more of those criteria comprised the social frailty group, whereas those with none or one criterion comprised the robust group, in accordance with the study by Makizako et al. [9].

#### 2.2.3. Assessment of Meaningful Activities

In this study, meaningful activities were operationally defined as ‘activities that individuals consider important in their daily life’ [20]. The Aid for Decision-making in Occupation Choice (ADOC) is an iPad application (Apple, Cupertino, CA, USA), that we used to gather data on meaningful activities [21]. ADOC was developed to identify activities meaningful to clients in rehabilitation [21]. The ADOC activities are used included in the International Classification of Functioning, Disability and Health (ICF) and consist of the following components: self-care, mobility, domestic life, work/education, interpersonal interaction, social life, sport, and leisure. Participants of Tarumizu Study were asked to select up to five meaningful activities from the ADOC face-to-face interviews and rated their satisfaction with each activity on a scale of 1–5 (1: very dissatisfied, 5: very satisfied). Previous assessments of the ADOC satisfaction measure have found it to be reliable and valid [22]. Participants rated their performance in each of the meaningful activities they had selected on a scale of 1–10 (1: with great difficulty, 10: perfectly). They were also asked how often they performed each meaningful activity (how many times a year, month or week). We then converted each of these frequencies into the number of times per 365 days for evaluation. The researchers in this study were occupational therapists and occupational therapy students. Before the study, we conducted two lectures, of around 2 h each, on methods for investigating meaningful activities. In addition, on the day of the study, we provided approximately 30 min of practical training to occupational therapists and occupational therapy students before the survey was administered. The study was conducted 24 times over the course of a year, and the researchers provided the above educational material each time. We examined the activities that participants considered the most meaningful.

#### 2.2.4. Other Variables

Data on sociodemographic and clinical variables, including age (years), sex, marital status, education (years), medications (n/day), history of hypertension, cognitive function, physical frailty, higher-level competence and depressive symptoms, were collected. Licensed doctors or nurses measured the resting blood pressure of participants at the study site. Education, medical history and medication data were obtained during face-to-face interviews. Cognitive function was assessed using the National Centre for Geriatrics and Gerontology-Functional Assessment Tool (NCGG-FAT). This measures four cognitive domains: memory, attention, executive function, and processing speed [23]. A score below 1.5 standard deviations from the average for a given domain, allowing for a participant’s age and level of education, was defined as poor cognitive functioning in the given domain [24]. The NCGG-FAT has been reported to be highly reliable and valid compared to traditional neurocognitive scales [23]. The NCGG-FAT has been used in previous studies with community-dwelling older adults [8,25,26,27]. Physical frailty was defined using the five components suggested by Fried: slowness, weakness, exhaustion, low physical activity and weight loss [4]. Participants with three or more of these components were classed as physically frailty. Higher-level competence was assessed using the Japan Science and Technology Agency Index of Competence (JST-IC), assesses higher levels of instrumental activities of daily living (IADL) within the context of modern society [28]. This instrument consists of the following four subscales (possible score range for each subscale: 0–4): Technology usage, Information practice, Life management, and Social engagement. Higher total scores reflect greater competence on higher-level IADLs (total possible score range: 0–16). Depressive symptoms were assessed using the 15-item Geriatric Depression Scale (GDS15), with a total score of ≥5 classed as depression [29,30,31].

### 2.3. Statistical Analysis

Statistical analyses were performed to compare the characteristics of community-dwelling older adults with and without heart disease. Data were analyzed using student’s *t*-tests for continuous variables, Mann–Whitney U tests for ordinal variables and Pearson’s χ2 tests for categorical variables. The relationship between heart disease and social frailty was examined by logistic regression analyses. There were three regression models: a crude model, adjusted model 1 and adjusted model 2. The presence of social frailty was set as the dependent variable in all models. In the crude model, the presence of heart disease was set as the independent variable. Adjusted model 1 was adjusted using demographic variables (age, sex, marital status, education) as covariates, whereas adjusted model 2 was adjusted using demographic variables, history of hypertension, medications, poor cognitive status, physical frailty, higher-level competence and depressive symptoms as covariates. Pearson’s χ2 test was used to compare the five aspects of social frailty addressed by the five questions on the social frailty measure between the with and without heart disease groups. Effect size was estimated using Cramer’s V, Cohen’s d, and r to determine the degree of difference.

The number of meaningful activity categories selected by individuals with and without heart disease were compared using Pearson’s χ2 test, Mann–Whitney U test for satisfaction and performance, and student’s t-tests for frequency.

All statistical analyses were performed using the IBM SPSS Statistics version 26.0 (IBM Corp., Armonk, NY, USA). A *p*-value of <0.05 was considered statistically significant.

## 3. Results

### 3.1. Participant Characteristics

The demographic and clinical characteristics of the study participants are shown in Table 1. The 630 participants were divided into two groups: 79 were those with heart disease (12.5%) and 551 were those without heart disease (87.5%). The heart disease group contained a significantly higher percentage of female participants (*p* = 0.004), and its members were significantly older (*p* < 0.001), had lower higher-level competence (*p* < 0.001), a higher rate of physical frailty (*p* = 0.011), social frailty (*p* = 0.002), poor cognitive functions (*p* = 0.018) than those in the group without heart disease.

### 3.2. Association between Heart Disease and Social Frailty

The logistic regression models showing the relationship between heart disease and social frailty are presented in Table 2. The unadjusted (crude) model showed a significant association between heart disease and social frailty [OR, 2.48; 95% CI, 1.47–4.17; *p* = 0.001)]. After adjusting for potential covariates, heart disease has remained significantly related to social frailty [adjusted model 1: OR, 2.20; 95% CI, 1.28–3.74; *p* = 0.004; adjusted model 2: OR, 1.97; 95% CI, 1.06–3.67; *p* = 0.032].

### 3.3. Association between Heart Disease and the Five Elements of Social Frailty

The relationships between heart disease and the five aspects of social frailty measured by the social frailty scale are presented in Table 3. Those with heart disease were significantly more likely to have ‘not feeling helpful to friends or family’ (*p* = 0.010) and to ‘not talking with someone every day’ (*p* = 0.041) than those without heart disease.

### 3.4. Characteristics of Meaningful Activities in Those with and without Heart Disease

Table 4 shows characteristics of meaningful activities identified by those with and without heart disease. There were no significant differences in category, satisfaction with, or performance, meaningful activities between the two groups. However, the frequency of meaningful activities was significantly lower in those with heart disease than in those without (*p* = 0.041). The frequency of both groups’ participation in meaningful activities by category is shown in Table 5. Among the activities selected, participants with heart disease engaged in self-care (*p* = 0.020) and interpersonal interaction (*p* = 0.019) significantly less frequently than those without.

## 4. Discussion

The main findings of this study were as follows: (1) heart disease is independently associated with social frailty in community-dwelling older adults, and (2) community-dwelling older adults with heart disease engaged in meaningful activities significantly less frequently than those without heart disease. To the best of our knowledge, this is the first study to examine the relationship between heart disease and social frailty and the characteristics of the meaningful activities of community-dwelling older adults with heart disease.

### 4.1. Heart Disease and Social Frailty in Community-Dwelling Older Adults

Among the 79 (12.5%) participants in this study with heart disease, 26 (32.9%) met the criteria for social frailty. This was a higher rate than that seen in previous studies (10.2–20.5%) that have used a similar operational definition of social frailty in community-dwelling older adults [9,32,33]. In addition, heart disease was significantly associated with social frailty in a logistic model adjusted for covariates. Hence, community-dwelling older adults with heart disease were more likely to suffer from social frailty than those without heart disease. Previous studies have found that loneliness and a lack of social support are associated with a poor prognosis after the onset of heart disease [34]. Social participation is likely to reduce loneliness, increase social support and maintain social roles, networks and activities among older adults with heart disease [35]. All aspects of social frailty were higher in our study than in previous research with community-dwelling older adults [9]. In particular, older adults with heart disease were more likely to experience ‘not feeling helpful to friends or family’ and ‘not talking with someone every day’ than those without heart disease. Even among older adults with chronic pain, ‘not feeling helpful to friends or family’ was common response [36]. The lack of a clear social role that provides a sense that one is contributing to the community, family or a particular group is a significant predictor of functional decline among community-dwelling older adults [37]. Therefore, engaging older adults with heart disease in activities that allow them to feel useful and valuable is vital, particularly in those with decreased physical function, or cognitive functions. ‘Not talking with someone everyday’ corresponds to a lack of social support [38]. Older adults with heart disease may interact less with others and have less access to social support as a result of reduced exercise tolerance and physical issues that impede engagement in many social activities. However, reduced interaction and social support can lead to depression and a more rapid decline in physical and cognitive health [39]. In patients with heart disease, advanced IADL skills such as disease and medication management are important, and there is a greater need for social support [40]. Social engagement may play an important role in allowing older adults with heart disease to maintain their independence and continue living in the community. Increasing social participation in older adults with heart disease can have positive effects on their physical and mental health, maintain or improve their quality of life and reduce the likelihood of rehospitalization [11].

### 4.2. Characteristics of Meaningful Activities of Community-Dwelling Older Adults with Heart Disease

In this study, there were no significant differences in satisfaction with performance of meaningful activities in community-dwelling older adults with and without heart disease, but those with heart disease engaged in these activities significantly less frequently. Allowing for the fact that different activities involve different frequencies of engagement, these results underscore the importance of frequent social engagement through meaningful activities for older adults with heart disease. Meaningful activities provide a sense of purpose and increase life satisfaction for older adult [41]. In recent years, the number of older adults who continue to work for purpose in life and participate in social activities through volunteer work and senior citizens’ clubs is on the rise. Engaging in activities of one’s own choice can reduce the incidence of frailty and dementia and maintain physical functions [42]. Thus, it is conceivable that performing meaningful activities may have a positive impact on the physical and mental functioning of the older adults as well as their social participation. These reports may also hold true for patients with heart disease. A previous study found that older adults with heart disease who continued to work before admission reduced social frailty and improved prognosis [15]. This showed that it is possible that heart disease patients may participate in society through activities such as work. However, activity can be limited in some patients with heart disease due to reduced exercise tolerance and risk management [43]. Therefore, engagement in meaningful activities at an increasing frequency that consider that risk management and exercise tolerance may effectively promote social participation. By category, those with heart disease performed self-care and interpersonal interaction significantly less frequently than in those without. People with heart disease tend to have a lower capacity for self-care ability, yet require more complex health management, including maintenance of a medication regimen, dietary modifications and weight control [44]. Social participation, including interpersonal interaction, has been reported to be effective in creating an environment for receiving social support such as self-care [34]. Social participation facilitates both social support and self-care and thereby may reduce the risk of rehospitalization.

However, this study has several limitations. First, heart disease was ascertained from pre-existing self-reported data. This may have led to an underrepresentation of the proportion of heart disease (12.5%) in the general population of older adults and to the large difference in the size of the groups of those with and without heart disease. However, the prevalence of heart disease in this study was at a reasonable rate among the general community-dwelling older adults. Secondly, the cross-sectional design prevented inference of a causal relationship between heart disease and social frailty. Future longitudinal studies are needed to establish this relationship. Third, Makizako’s social frailty scale items have not been validated in heart disease patients. However, the measurements of social frailty in this study were all clinically applicable and are simple, inexpensive, and are increasingly used in patients with heart disease. Nevertheless, these instruments should be externally validated in future studies. Fourth, the lack of data on factors such as poverty and deprivation, which are potential interactions of social frailty. A more detailed investigation of these points is needed in the future. Fifth, we did not investigate the degree of frequency in this study, and it is difficult to examine each category of activity. In the future, we believe it is necessary to investigate in detail not only the frequency of meaningful activities, but also the degree and each category of activity. Sixth, the participants in this study were community-dwelling residents undergoing a health check-up, so they were not selected randomly. Only those able to participate in the health check-up were included, and the number of participants was not determined based on sample size calculations. Finally, the data were from a single city, and social frailty may be affected by cultural and regional differences. Further studies in other regions are needed to consider regional characteristics. However, despite these limitations, this study is still useful in identifying the association between heart disease and social frailty in community-dwelling older adults and the characteristics of meaningful activities among community-dwelling older adults with heart disease.

## 5. Conclusions

This study found that community-dwelling older adults with heart disease may be more likely to have social frailty. Since this demographic is inclined to engage in meaningful activities less frequently, they should be supported and encouraged to engage in such activities on a regular basis. Promoting social participation through meaningful activities may reduce re-admissions and maintain the quality of life of older adults with heart disease. However, since this was a cross-sectional study, further longitudinal research is needed to support our findings and establish a causative relationship.

## Figures and Tables

**Figure 1 ijerph-19-15167-f001:**
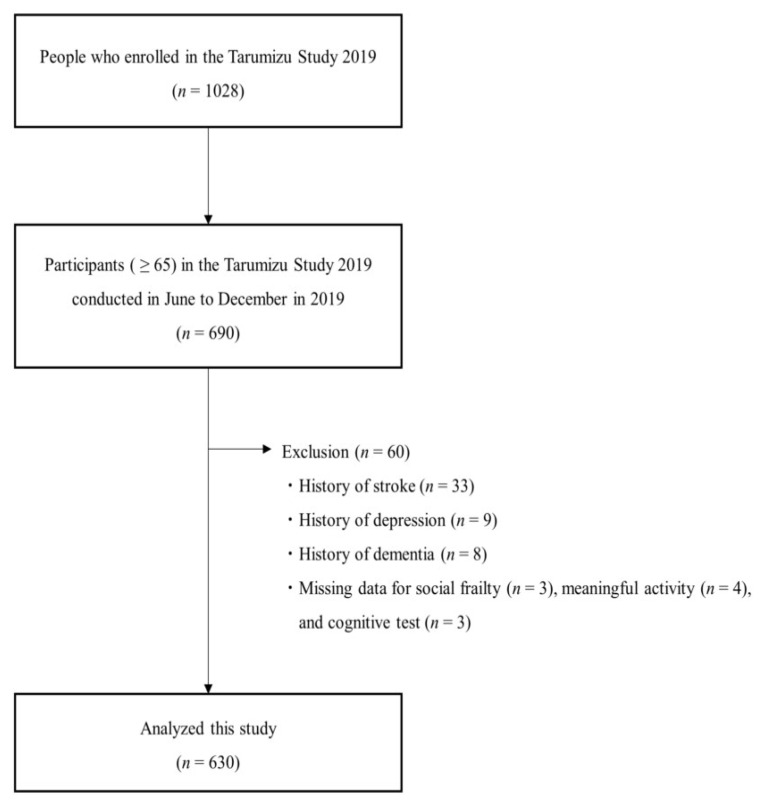
Flowchart for study inclusion and exclusion.

**Table 1 ijerph-19-15167-t001:** Participant Characteristics.

	With Heart Disease (n = 79)	Without Heart Disease (n = 551)	*p*-Value
Age, mean ± SD (years)	76.3 ± 6.4	73.9 ± 6.3	<0.001 ^a^
Female, n (%)	38 (48.1%)	358 (65.0%)	0.004 ^b^
Medications, mean ± SD (numbers)	6.5 ± 3.7	2.9 ± 2.8	<0.001 ^b^
Married, n (%)	74 (93.7%)	530 (96.2%)	0.293 ^a^
Educational history, mean ± SD (years)	11.1 ± 2.1	11.5 ± 2.2	0.238 ^a^
History of hypertension, n (%)	37 (46.8%)	269 (48.8%)	0.810 ^a^
Poor cognitive status, n (%)	25 (31.6%)	110 (20.0%)	0.018 ^b^
Physical frailty, n (%)	11 (13.9%)	37 (6.7%)	0.011 ^b^
slowness, n (%)	12 (15.2%)	48 (8.7%)	0.067 ^b^
weakness, n (%)	21 (26.6%)	70 (12.7%)	<0.001 ^b^
exhaustion, n (%)	17 (21.5%)	99 (18.0%)	0.446 ^b^
low physical activity, n (%)	15 (19.0%)	88 (16.0%)	0.498 ^b^
weight loss, n (%)	17 (21.5%)	99 (18.0%)	0.446 ^b^
Higher-level competence, mean ± SD (points)	10.6 ± 3.2	12.0 ± 3.0	<0.001 ^a^
Social frailty, n (%)	26 (32.9%)	91 (16.5%)	0.002 ^b^

SD, standard deviation, ^a^ Student’s *t*-test, ^b^ Person’s χ2 test.

**Table 2 ijerph-19-15167-t002:** Association Between Heart Disease and Social Frailty.

	Crude Model			Adjusted Model 1			Adjusted Model 2		
	OR	95% CI	*p*- Value	OR	95% CI	*p*- Value	OR	95% CI	*p*- Value
Heart disease	2.48	1.47–4.17	0.001	2.20	1.28–3.74	0.004	2.21	1.01–4.83	0.047

OR, odds ratio; CI, confidence interval; In each model, the presence of social frailty was set as the dependent variable; Crude model: the presence of heart disease was set as the independent variables; Adjusted model 1: the presence of heart disease was set as independent variables, and adjusted using demographic variables as covariates; Adjusted model 2: the presence of heart disease was set as the independent variables, and adjusted using demographic variables, history of hypertension, medications, cognitive status, physical frailty, higher-level competence and depressive symptoms as covariates.

**Table 3 ijerph-19-15167-t003:** Association Between Heart Disease and the Five Elements of Social Frailty.

	With Heart Disease (n = 79)	Without Heart Disease (n = 551)	ES	*p*-Value ^a^
going out less frequently compared with last year (yes), n (%)	18 (22.8%)	82 (14.9%)	0.07	0.072
not visiting friends sometimes (yes), n (%)	20 (25.3%)	102 (18.5%)	0.06	0.152
not feeling helpful to friends or family (yes), n (%)	11 (13.9%)	33 (6.0%)	0.10	0.010
living alone (yes), n (%)	26 (32.9%)	152 (27.6%)	0.04	0.326
not talking with someone every day (yes), n (%)	13 (16.5%)	50 (9.1%)	0.08	0.041

ES, effect size; ^a^ Person’s χ^2^ test.

**Table 4 ijerph-19-15167-t004:** Meaningful Activities in Those with and Without Heart Disease.

	With Heart Disease (n = 79)	Without Heart Disease (n = 551)	ES	*p*-Value
Category of meaningful activities, n (%)			0.09	0.549 ^b^
self-care	6 (7.6%)	70 (12.7%)		
mobility	0 (0%)	6 (1.1%)		
domestic life	12 (15.2%)	85 (15.4%)		
work/education	6 (7.6%)	31 (5.6%)		
interpersonal interaction	14 (17.7%)	93 (16.9%)		
social life	4 (5.1%)	54 (9.8%)		
sport	10 (12.7%)	64 (11.6%)		
leisure	27 (34.2%)	148 (26.9%)		
Satisfaction of meaningful activities, Median (IQR)	5 (4–5)	5 (4–5)	0.01	0.847 ^c^
Performance of meaningful activities, Median (IQR)	9 (7–10)	9 (7–10)	0.03	0.452 ^c^
Frequency of meaningful activities, mean ± SD (days)	212.2 ± 147.3	247.7 ± 143.9	0.25	0.041 ^a^

IQR, interquartile range; ^a^ Student’s *t*-test, ^b^ Person’s χ^2^ test, ^c^ Mann-Whiney U test.

**Table 5 ijerph-19-15167-t005:** Frequency of Meaningful Activities by Category.

	With Heart Disease (n = 79)	Without Heart Disease (n = 551)	ES	*p*-Value
self-care, mean ± SD (days)	260.5 ± 143.8 (n = 6)	343.4 ± 75.6 (n = 70)	0.91	0.020 ^a^
mobility, mean ± SD (days)	(n = 0)	330.0 ± 54.2 (n = 6)		
domestic life, mean ± SD (days)	318.2 ± 113.6 (n = 12)	325.2 ± 101.7 (n = 85)	0.07	0.826 ^a^
work/education, mean ± SD (days)	234.3 ± 130.4 (n = 6)	260.6 ± 115.8 (n = 31)	0.22	0.621 ^a^
interpersonal interaction, mean ± SD (days)	112.8 ± 111.7 (n = 14)	214.5 ± 153.7 (n = 93)	0.68	0.019 ^a^
social life, mean ± SD (days)	129.3 ± 161.6 (n = 4)	138.3 ± 144.1 (n = 54)	0.06	0.905 ^a^
sport, mean ± SD (days)	170.8 ± 121.5 (n = 10)	192.2 ± 136.6 (n = 64)	0.16	0.642 ^a^
leisure, mean ± SD (days)	228.6 ± 157.9 (n = 27)	236.6 ± 145.2 (n = 148)	0.05	0.796 ^a^

^a^ Student’s *t*-test; Conversion method: we asked participants how often they performed each meaningful activity (how many times a year, month, or week), and then converted each of these frequencies into the number of times per 365 days for evaluation.

## Data Availability

The data presented in this study are available on request from the corresponding author. The data are not publicly available due to privacy and ethical restrictions.

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
