# Peer review of "Social Frailty and Meaningful Activities among Community-Dwelling Older Adults with Heart Disease"

_ijerph, 2022, doi:10.3390/ijerph192215167_

Round 1

Reviewer 1 Report

1. Please show the Effect size in tables 4 and 5.

2. When we evaluate an activity using ADOC, even if it has a satisfaction rating of 1, we consider it meaningful (because we evaluate meaningful activities with the ADOC). Does this mean it is crucial to increase the frequency of meaningful activities even with such a low satisfaction level? Please consider the degree and frequency of meaningful activities. Following the results, it could be emphasized that the heart disease group had higher satisfaction and performance.

Reviewer 2 Report

I thank the authors to provide this study about a rarely discussed topic.

Here are my comments:

-         Introduction:

Authors should better define social frailty, which is not so clear as physical frailty. Furthermore, the rational for studying meaningful activities should be improved. Please add some reference for the sentence « Physical inactivity in patients with cardiac disease may also affect meaningful activity » lines 63-64.

-         Material & Methods:

·        Why an age cut-off at 65 years? Especially in Japan

·        Concerning the analyzed data:

o   I think the authors might have include some poverty or deprivation Index, which could influence IADLs

o   Same remark for presence for home help or not

o   Why choose a non-international tool (e.g. MMSE or MoCA) for cognitive assessment?

o   Why include BP levels in the models? Maybe not pertinent

o   Some physical evaluation like walk speed would have been relevant, maybe the authors could extract the data from “the physical frailty” item and analyze it independently

-         Discussion

If the discussion is well conducted, I would have seen more limitations regarding the lack of social data (as said before, home help etc…) and the potential interaction between asthenia (or lower walk speed) with social frailty which could be more relevant for a geriatrician reader.

Minor English checking (frail and not frailty line 56, flowchart)

I suggest a MAJOR REVISION for the elements mentioned above. If the authors cannot provide some additional analyzes, I would appreciate sufficient justifications for this.

Round 2

Reviewer 2 Report

ok